# Ionized acrylamide-based copolymer / terpolymer hydrogels for recovery of positive and negative heavy metal ions

Kentaro Fujimoto[1☯¤], Brian Adala Omondi[2,3☯], Shinya Kawano[2,3‡], Yoshiki Hidaka[2,3‡], Kenji Ishida[2,3‡], Hirotaka Okabe[2,3☯]*, Kazuhiro Hara[2,3☯]

**1** Department of Applied Quantum Physics and Nuclear Engineering, Graduate School of Engineering, Kyushu University, Fukuoka, Japan, **2** Department of Applied Quantum Physics and Nuclear Engineering, Faculty of Engineering, Kyushu University, Fukuoka, Japan, **3** Center for Research and Education of Environmental Technology, Kyushu University, Fukuoka, Japan

☯ These authors contributed equally to this work.
¤ Current address: Sumitomo Riko Company Limited, Komaki, Japan
‡ SK, YH and KI also contributed equally to this work.
* okabe@ap.kyushu-u.ac.jp

**Data Availability Statement:** All relevant data are within the paper and its Supporting Information files.

## Abstract

In this study, we explored the effective capture of both cations and anions onto a single adsorbent. Acrylamide (AAm) served as the polymer backbone, onto which co-monomers sodium p-styrenesulfonate (SS) and N,N-dimethylaminopropyl acrylamide (DMAPAA) were grafted, creating ionized polymer hydrogel adsorbents. These adsorbents were engineered for the synergistic separation and recovery of heavy metal cations and anions from concentrated solutions, focusing specifically on industrially significant ions such as $Ni^{2+-}$, $Cu^{2+}$, $Zn^{2+}$ and $(Cr_2O_7)^{2-}$. The adsorption and desorption behaviors of the AAm terpolymer hydrogels were investigated across various pH solutions, considering the competition and concentrations of these specific metal ions. Moreover, the study delved into the effects of the internal pH environment within the hydrogel adsorbents, determining its impact on the type of metal adsorbed and the adsorption capacity. Our findings indicated that the adsorption of cations was enhanced with a higher proportion of SS relative to DMAPAA in the hydrogel. In contrast, significant anion capture occurred when the concentration of DMAPAA exceeded that of SS. However, equal ratios of SS and DMAPAA led to a noticeable reduction in the adsorption of both types of substrates, attributed to the counteractive nature of these co-monomers. To enhance the adsorption efficiency, it may be necessary to consider methods for micro-scale separation of the two types of monomers. Additionally, the adsorption capacity was observed to be directly proportional to the swelling capacity of the hydrogels. For complete desorption and separation of the cations and anions from the adsorbent, the application of concentrated NaOH solutions followed by $HNO_3$ was found to be essential. Given the varying concentrations of cation and anion pollutants, often present in heavy metal factory effluents, it is crucial to fine-tune the ratios of DMAPAA and SS during the synthesis process. This adjustment ensures optimized efficiency in the decontamination and recovery of these significant heavy metal ions.

**Funding:** This work was supported by JSPS KAKENHI Grant Numbers JP21656239, JP24360398, JP19H02660, JP21H01871. The funders had no role in study design, data collection and analysis, decision to publish, or preparation of the manuscript.

**Competing interests:** The authors have declared that no competing interests exist.

# Introduction

The search for alternative technology for heavy metal decontamination is an on-going endeavor that seeks to achieve resource sustainability in heavy metal industries. An example of the toxic effects of heavy metal pollution on human and aquatic life is the Minamata disease, which occurred in the Minamata Bay area of Japan in 1956 to 1965, due to methylmercury poisoning [1, 2]. Since then, stringent rules have been enforced worldwide on the industrial wastewater handling and disposal practices. For many years until presently, conventional method of heavy metal wastewater treatment has relied on the coagulation sedimentation technique [3, 4] whereby heavy metal in wastewater is converted into insoluble metal hydroxides, which then sediments as a sludge cake. Both the supernatant and the precipitated cake is often discarded due to low-quality treatment technique. This has created a persistent burden on industries sustainability, since the cake mass requires secondary disposal, and 95% the treated water is discarded, therefore there is no resource recovery nor recycling. At the same time, there is demand for large volumes of fresh water every time, which is unsustainable in the long term. Several researches have attempted to bridge this gap using methods such as membrane technology [5], magnetic nanoparticles functionalized polymers [6, 7], the use of zeolite-based absorbents [8], and carbon nanotubes [9].

Methodologies for heavy metal wastewater decontamination requires a deliberate design of functional groups in order to effectively eliminate the metals from the water in a way that would facilitate recovery of these metals while ensuring clean recycle-quality treated water [10, 11]. This warrants a keen analysis of the wastewater conditions in their actual industrial environment such that the new technology design and feasibility could be accurately assessed. Heavy metal wastewaters are usually characterized as multi-contaminated with several metal ions (inorganic), including organic contaminants. Among the metal ions (which are mainly positively charged cations) there also exists the anionic hexavalent chromium $(Cr_2O_7)^{2-}$. Treatment techniques should thus be designed to simultaneously remove both cations and anions.

Cr(VI) is a notorious pollutant due to its high toxicity and prevalence in various heavy metal industries such as the electroplating industries [12]. The conventional handling of this pollutant is reduction from its hexavalent to the mildly toxic trivalent form, and subsequent precipitation into the Cr(III) hydroxide using coagulating agents. Traditional reduction treatments utilize the sulfur compounds such as sodium bisulfite or ferrous sulfate. But these methods unsuitability includes: (i) generation of large amount of secondary waste that would require additional disposal, and (ii) the excessive large volumes of coagulating agents that would be required to achieve total coagulation/precipitation. Evidently, new and efficient technology is desired that would achieve this pollutant removal: (a) directly without the pre-requisite for reduction, (b) enabling possible recovery of the Cr metal following the decontamination method, and (c) guarantee feasibility even at low concentrations of Cr(VI). Absorption has gradually emerged as the most probable technique based on these criteria. Absorbents such as iron nanoparticles, activated carbons have been consistently investigated with various outcomes [13–16]. Challenges are prevalent, such as the one faced by Ponder et al. when their development of zero-valent iron nanoparticle absorbent could only achieve limited reduction of Cr(VI) to its trivalent form, rather than absorption [17]. Similarly, activated carbons gained accelerated popularity, but their development stagnated due to consistently low efficiency, such as Karthikeyan et al using *Hevea Brasiliensis* sawdust activated carbon managing only 20% extraction of total chromium from concentrated solutions [18]. This is because Cr(VI) is an unconventional metal anion, therefore specially dedicated functional groups and absorbent design is required to guarantee high absorption performance. In 2006, Hara et al were the first to pioneer absorbents capable of heavy metal-anion adsorptive

removal from concentrated solutions [19]. Recently in 2018, Jain et al by using $Fe_2O_3$-functionalized activated carbons, managed 85% Cr(VI) extraction from concentrated solutions, via electrostatic attraction/repulsion mechanism [20]. It was the amino/hydroxyl functional groups that were mainly responsible for the Cr(VI) ions extractive capture.

Building on these foundations, this work now attempts multiple ions capture (selected heavy metal ions, Ni(II), Cu(II), Zn(II) and Cr(VI) onto a single absorbent framework. Hydrogels was the choice absorbent owing to its remarkable flexibility in design and ability to incorporate multiple functional groups while retaining the low-cost nature; and ability to be regenerated for reuse [21–23]. Successful applications in dilute concentration environments coupled with hydrogels ability to imbibe water and swell is also a unique property of hydrogel absorbents that facilitates higher absorption capacities [24, 25]. For this study, a multifunctional terpolymer hydrogel was designed with dedicated monomer functional groups for Cr (VI) capture and a separate functional group for heavy metal cations absorption via electrostatic attraction/repulsion mechanism in both cases.

In the pursuit to achieve simultaneous extraction of both the heavy metal-anion and cation within a single absorbent, hydrogel's polymeric network offers great versatility by enabling multiple functional groups to be attached to the main polymer backbone [23]. Cr(VI) has an unusual property occurring as metal anion $(Cr_2O_7)^{2-}$) instead of cation. Consequently, the main obstacle has persisted of how to ensure heavy metal anion capture of Cr(VI) directly onto an absorbent, without any intermediate steps such as using reducing agents to modify the Cr(VI) oxidation state. To achieve this, earlier works by our research group focused on studying and gradually modifying the type of grafted functional group onto AAm based hydrogels, and analyzing its Cr(VI)-capturing behavior. Initially, AAm hydrogel functionalized with dimethylaminoethylmethacrylate and methyl chloride (DMAEMA-MeCl/AAm) showed higher anion purifying efficiency than any Cr(VI)-capturing absorbent during that time. Later, we attempted to modify the network configuration of DMAEMA monomer by eliminating the methyl-base, to instead use dimethylaminoethylacrylate (i.e. DMAEA-MeCl/AAm gel). The performance of the gel was remarkably higher and this results informed to eliminate alkyl group appendages from the main chain [26, 27]. Presently, this study further modified the DMAEA chain responsible for Cr(VI) absorption, by increasing the backbone from ethyl to propyl chain, and replaced the end group from acrylate to amide moiety by using a new monomer diaminopropyl acrylamide (DMAPAA). DMAPAA use was expected to demonstrate significantly higher absorption of heavy metal anion Cr(VI). Further, to facilitate simultaneous cation capture, an ionized functional monomer: styrene sulfonic acid (SS) was also grafted onto the main acrylamide chain to form the final DMAPAA-SS/AAm hydrogel absorbent in this study. Ionization of each group occurs as the following Eqs 1 and 2 [28].

$$R - SO_3H \rightarrow R - SO_3^- + H^+ \tag{1}$$

$$C_7H_{14}N_2O + H_2O \rightarrow C_7H_{15}N_2O^+ + OH^- \tag{2}$$

## Material and methods

### Materials

Acrylamide (AAm), sodium p-styrene sulfonate (SS) and *N,N'*-dimethylaminopropylacrylamide (DMAPAA) monomer reagents, including polymerization reagents ammonium persulfate (APS) and *N,N'*-Methylenebisacrylamide (NMBA) (Fig 1) were obtained from Merck (Sigma Aldrich), Germany. All reagents were of high analytical grade and were used as received without further purification. Similarly, standard heavy metal solutions (high purity, ICP-MS grade) of "chromium (VI) prepared with $(NH_4)_2Cr_2O_7$", copper, nickel and zinc

**Fig 1. Chemical formulas of reagents used in this study.**

diluted in $HNO_3$ were sourced from Merck, and these used in the experiments by diluting with water. All references to water shall refer to the ultrapure deionized water purified by Puric-Omega by Organo Co., Japan.

The abbreviations used in the text are shown in parentheses below the reagent names.

## Sample preparation

AAm-based copolymer and terpolymer hydrogels as shown in Fig 2 were prepared in combination with SS and/or DMAPAA co-monomers. SS and DMAPAA monomers selection targeted their respective sulfonate and ammonium functional groups, which will trigger absorption of cations and anions via electrostatic interactions.

In the synthesis recipe summarized in Table 1, concentration of AAm was maintained at 500 mM, whereas the SS and DMAPAA concentrations alternated to ensure sum of 700 mM in all cases. Additionally, for each sample, the volume of water was adjusted such that the sum of all constituents, including the polymerization reagents, was 50 g. Finally, successful hydrogelation was undertaken in a thermostatic bath at 50˚C for 24 hours.

The yield molecular weight for each type of terpolymer hydrogel obtained was computed by factoring the respective molar ratio of each contributing monomer, and the results are summarized in Table 1.

In a similar way, copolymer hydrogels (using only AAm and DMAPAA) were prepared, with alternating concentrations of each monomer as summarized in Table 2. These two classes of hydrogels (terpolymer and copolymer hydrogels) will be used for subsequent absorption investigations.

## Structural analysis using NMR

In the spectra in S1 Fig obtained by preliminary FT-IR measurements, changes were observed in S = O stretching derived from SS and amide band II derived from DM, but changes of SS and DM overlapped with the one due to AAm. So, it was difficult to conclude that these had been introduced. Therefore, the $^{13}$C NMR was used to confirm that the desired functional

**Fig 2. Chemical formulas of the synthesized gels.**

groups were incorporated into the prepared hydrogels. The JNM-ECA400 FT NMR spectrometer (JEOL Ltd., Japan) was used for all NMR analysis. The samples used for NMR analysis [SS0-DM0, SS200-DM0, SS0-DM200, SS100-DM100] were initially dried until constant dry weight, then crushed into fine powdery form prior to analysis as shown in S2 Fig. All original NMR data with measuring condition are shown in Supporting information.

## Metal absorption experiments

**Adsorptive potential and behavior of (AAm-SS-DMAPAA) terpolymer hydrogels.** The experimental framework is summarized in Fig 3. Using the samples in Table 1, the hydrogels were sliced into 5 mm square cubes (0.25 g) and washed in excess water for 24 hours to elute any unreacted reagents or oligomers during gelation process.

**Table 1. Synthesis ratios for (AAm–SS–DMAPAA) terpolymer hydrogels.**

| Name | Solvent (mL) | AAm (mM) | SS (mM) | DMAPAA (mM) | NMBA (mM) | APS (mM) | Polymer MW |
|---|---|---|---|---|---|---|---|
| SS0 –DM0 | 47.60 | 700 | 0 | 0 | 12.3 | 2.5 | 19900 |
| (AAm only) | | | | | | | |
| SS0 –DM200 | 47.07 | 500 | 0 | 200 | 12.3 | 2.5 | 26700 |
| SS50 –DM150 | 46.53 | 500 | 50 | 150 | 12.3 | 2.5 | 27700 |
| SS100–DM100 | 46.50 | 500 | 100 | 100 | 12.3 | 2.5 | 28700 |
| SS150 –DM50 | 46.47 | 500 | 150 | 50 | 12.3 | 2.5 | 29700 |
| SS200 –DM0 | 47.00 | 500 | 200 | 0 | 12.3 | 2.5 | 30700 |

Meanwhile, heavy metal aqueous solution of Ni(II), Cu(II), Cr(VI) were prepared at initial concentrations of 5 mM each of different pH (pH adjusted to 1, 2, 3, 4 using $HNO_3$ and $NaOH_{(aq)}$).

In the absorption experiments, 0.25 g hydrogel absorbents were immersed into 10 mL of respective heavy metal solution (5 mM) for 72 hours at ambient conditions. Thereafter, the 7700x ICP-MS machine (Agilent, USA) was used to monitor the change in concentration of each solution (due to the action of the gel absorbent), and consequently to compute the absorption capacity for each hydrogel.

Separately, in order to corroborate the amount of metal ions that actually migrated from the solution into the absorbent network, the metal-laden absorbent was extracted and irradiated using Ethos One microwave digestion system (Milestone, Italy) to disintegrate the polymer chains so that it releases the embedded metal ions into nitric acid solution. The concentration of these released metal ions was then determined using ICP-MS system.

The flow of the experimental procedure and photos of the equipment used are shown.

**The effect of DMAPAA in (AAm–DMAPAA) copolymer hydrogels metal absorption.** The preliminary investigations using terpolymer hydrogels with higher DMAPAA than SS concentrations revealed high degree of Cr(VI) ions capture, in addition to Cu(II). Consequently, the authors sought to determine the influence of only DMAPAA in the absence of SS.

New set of hydrogels were prepared utilizing only AAm and DMAPAA monomers, with increasingly higher DMAPAA concentrations as depicted in Table 2. Thereafter, fresh copolymer hydrogels were sliced into 5 mm square cubes (0.25 g) and washed in ultrapure water for 24 hours. Before application on metal absorption, the authors sought to find out the initial condition of the hydrogel internal environment, by measuring the internal pH of each hydrogel type using a micro pH electrode probe (Orion™ 9810BN, Thermo Fisher Scientific, USA).

Batch absorption experiment was carried out by immersing 0.25 g gel fractions into 5mM combined multi-metal solutions comprising Cu(II), Ni(II), Zn(II) and Cr(VI) ions of pH 4. The effect of increasing DMAPAA concentrations, and the influence of initial pH environment of the hydrogel network, was analyzed against the metal-capture efficiency for each copolymer hydrogel type, in attempt to determine why DMAPAA hydrogels were able to absorb copper cations when it was only expected to absorb Cr(VI) anions.

**Hydrogel digestion and ICP-MS analysis.** In all cases during absorption, since the hydrogels extracted the metals from the solution into the hydrogel network during absorption process, the authors aimed to find out the actual quantity of metals transferred from solution and embedded into the respective hydrogel networks. To do this, the metal-absorbed hydrogels was inserted into ampoules containing 8 mL of 60% nitric acid. Sealed ampoules were irradiated with microwave radiation using Ethos One microwave digester system (Milestone, Italy). Irradiation ensured the polymer network of the hydrogels are disintegrated and the

**Table 2. Composition of (AAm–DMAPAA) copolymer hydrogels.**

| Name | Solvent (mL) | AAm (mM) | DMAPAA (mM) | NMBA (mM) | APS (mM) | Polymer MW |
|---|---|---|---|---|---|---|
| DM100 | 47.07 | 600 | 100 | 12.3 | 2.5 | 23300 |
| DM200 | 46.56 | 500 | 200 | 12.3 | 2.5 | 26700 |
| DM300 | 46.06 | 400 | 300 | 12.3 | 2.5 | 30120 |
| DM400 | 45.55 | 300 | 400 | 12.3 | 2.5 | 33520 |
| DM500 | 45.05 | 200 | 500 | 12.3 | 2.5 | 36930 |
| DM600 | 44.54 | 100 | 600 | 12.3 | 2.5 | 40300 |

The top two are copolymer hydrogels of AAm and SS or DM, and the bottom one is terpolymer hydrogel of AAm and SS and DM.

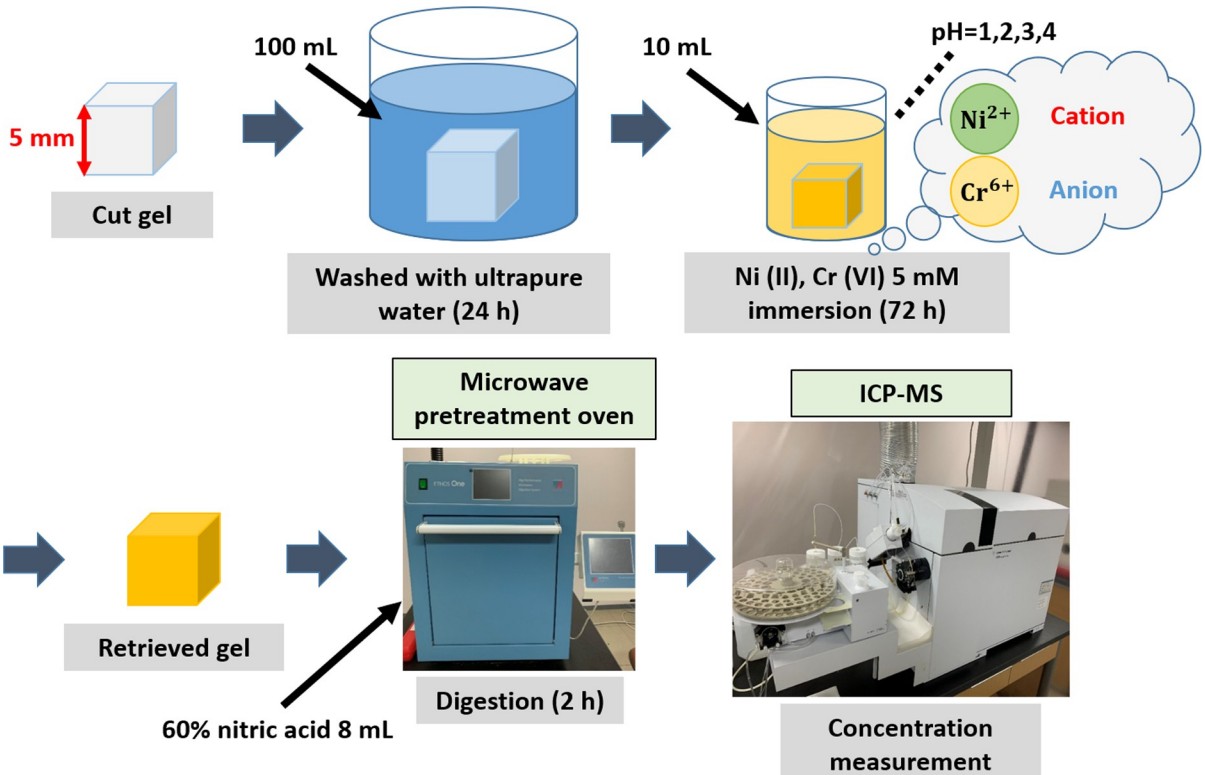

**Fig 3. Outline of heavy metal cation/-anion absorption experiments using (AAm–SS–DMAPAA) terpolymer hydrogel.**

liquefied hydrogel releases the absorbed metals into the acidic solution. The metal concentration of this acidic solution was analyzed using ICP-MS (Agilent 7700s, Japan) to find out the true absorption capacity of the hydrogels (Fig 3). The measured concentration (metals embedded in the absorbent) was compared with the initial data on change in concentration of heavy metal solutions during each absorption cycle.

**Hydrogel desorption experiments.** Following successful heavy metal absorption of positive and negative heavy metal ions and anions, the bipolar terpolymer hydrogels were desorbed (to regenerate the hydrogel absorbents), by immersing the hydrogels in concentrated acid and alkaline solutions. Initially, the metal-concentrated hydrogels were pretreated into 3M $HNO_3$ solution followed by 3M NaOH solution for 24 hours each, to selectively desorb the cation and anion metal substrates. Separately, the authors reversed the order of immersion (3M NaOH, followed by 3M $HNO_3$) in order to find out what will be the effect on type and degree of metal desorption. In all cases, the residual aqueous solution after desorption process was analyzed using ICP-MS in order to identify the type and quantity of metal ions desorbed from the absorbents.

## Results and discussion

### Structural analysis using NMR

The AAm only hydrogel (i.e. SS0-DM0 in Table 1) was used as the control sample. The $^{13}C$-NMR peaks of the terpolymer and copolymer hydrogels were compared to the control sample as shown in Fig 4 to identify the additional functional groups based on the carbon environments. The peaks in the spectra and the corresponding carbons in the chemical structures are

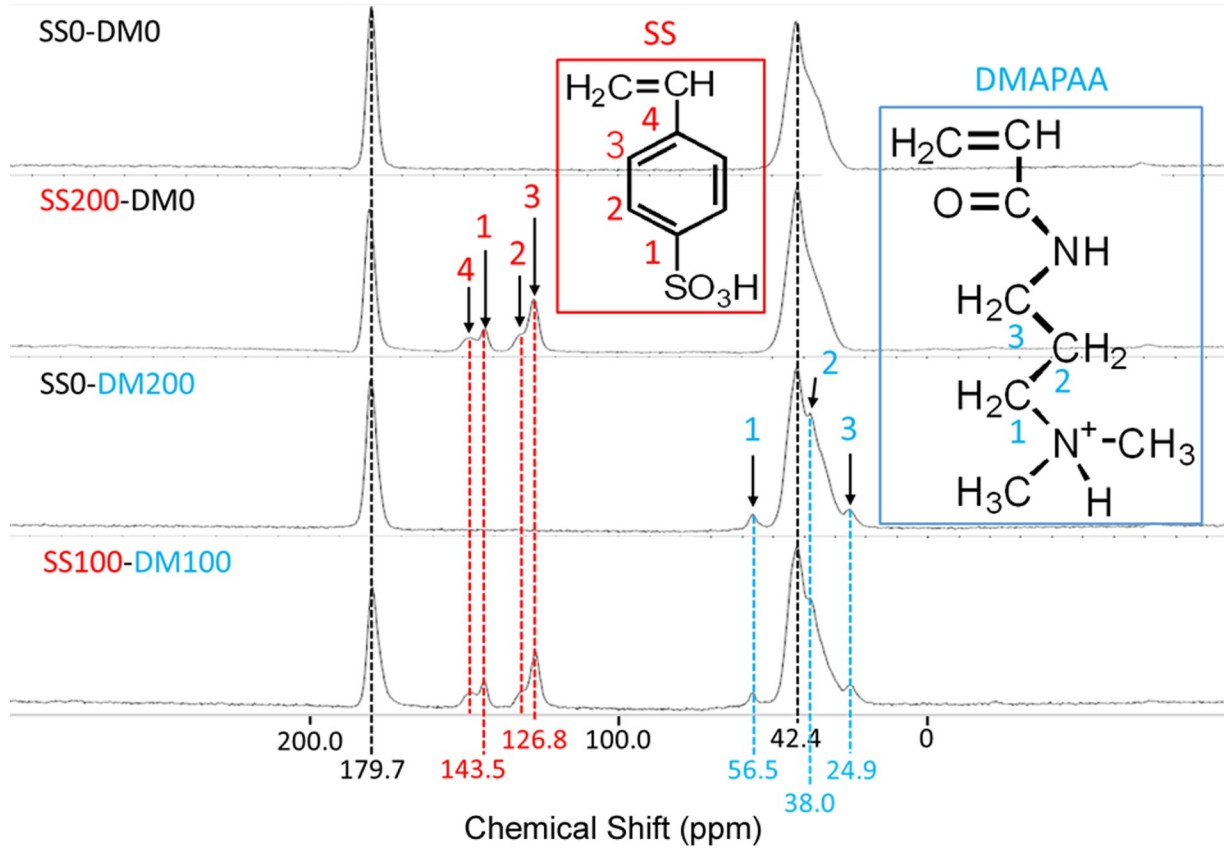

**Fig 4. Structural analysis of (AAm–SS–DMAPAA) hydrogel using $^{13}$C NMR spectroscopy.**

indicated by the same number. The peaks appearing around 42 and 180 ppm are observed in all samples, indicating that these peaks originate from the backbone AAm. The peak around 42 ppm is likely attributed to the alkyl carbon ($CH_2$- part) in AAm. In NMR spectroscopy, alkyl carbons generally exhibit peaks in the range of 30 to 50 ppm. For AAm, the methylene carbon in the vinyl group ($CH_2 = CH$-) corresponds to this range. The peak around 180 ppm is highly indicative of a carbonyl carbon (C = O). Carbonyl carbons usually show peaks in the range of 150 to 200 ppm. In AAm, the carbonyl group (-$CONH_2$) is expected to appear within this range. Four peaks were observed around 125 to 145 ppm in the gel containing SS. These are identified as the four carbon peaks that make up the benzene ring of styrene sulfonic acid. On the other hand, in the samples containing DMAPAA, three small side peaks at 24, 38 and 56 ppm were observed around the peak at 42 ppm of AAm. These can be identified derived from the three carbons of DMAPAA shown in Fig 4. According to the characteristic chemical shifts of the sulfonic acid group, and the influence of the amide group of DMAPAA, it was possible to confirm that indeed the target ionic groups (sulfonate group and quaternary ammonium) was introduced into the prepared hydrogels.

The red numbers show the chemical shifts derived from SS and the carbon atoms from which they originated, and the blue numbers show the chemical shifts derived from DMAPAA and the carbon atoms from which they originated. Peaks at 42 and 180 ppm, consistent across all samples, are attributed to AAm's alkyl and carbonyl carbons, respectively. The specific peaks related to styrene sulfonic acid in SS-containing gels and DMAPAA's carbons in its respective samples are identified.

### Terpolymer hydrogels absorption capacity and effect of swelling

**Absorption from combined heavy metal cation/anion solutions.** Absorption behavior of this new three-component absorbent was analyzed using two sets of cation/anion combined solutions, i.e. $Ni^{2+}/Cr^{6+}$ solution, and $Cu^{2+}/Cr^{6+}$ solutions of different pH. Against the $Ni^{2+}/Cr^{6+}$ solutions, the terpolymer hydrogel showed both high cationic and anionic-capturing affinities of metal ions depending on the ratio of either SS or DMAPAA functional group. As illustrated on Fig 5, the influence of pH on absorption behavior was significant but dependent on the amount of SS or DMAPAA in the hydrogel. In the control sample constituting only AAm monomer, the degree of absorption was low but equal for both Ni(II) and Cr(VI) capture at all pH levels, implying that AAm had the ability to remove both anion and cation simultaneously, albeit only slightly. As the amount of SS in the hydrogel network gradually increases relative to DMAPAA, the corresponding absorption of cation ($Ni^{2+}$) is higher and more prominent especially at pH 1–3. Whereas, if the amount of DMAPAA > SS in the hydrogel network, then more Cr(VI) is absorbed, more notably at pH 1–2. Overall, the maximum absorption of $Cr^{6+}$ is much higher that $Ni^{2+}$, implying the high efficiency of DMAPAA functional group on heavy metal-anion capture. Finally, if there is equal concentration of SS and DMAPAA in the hydrogel network, absorption was suppressed, occurring at lower efficiency than the control sample. Which informs that in order to effectively apply this absorbent, parity concentration of SS and DMAPAA is not desirable. Instead, since in all typical polluted environments the anion/cation concentration is never equal, therefore depending on the (measured) target solution concentrations of heavy metal anion/cation components, the synthesis design of the terpolymer hydrogel should be adjusted to utilize either higher SS or DMAPAA to guarantee high efficiency absorption.

Separately, when the type of cation in the combined solution was changed from $Ni^{2+}$ to $Cu^{2+}$, the absorption behavior significantly altered, as depicted in Fig 6. At slightly higher SS than

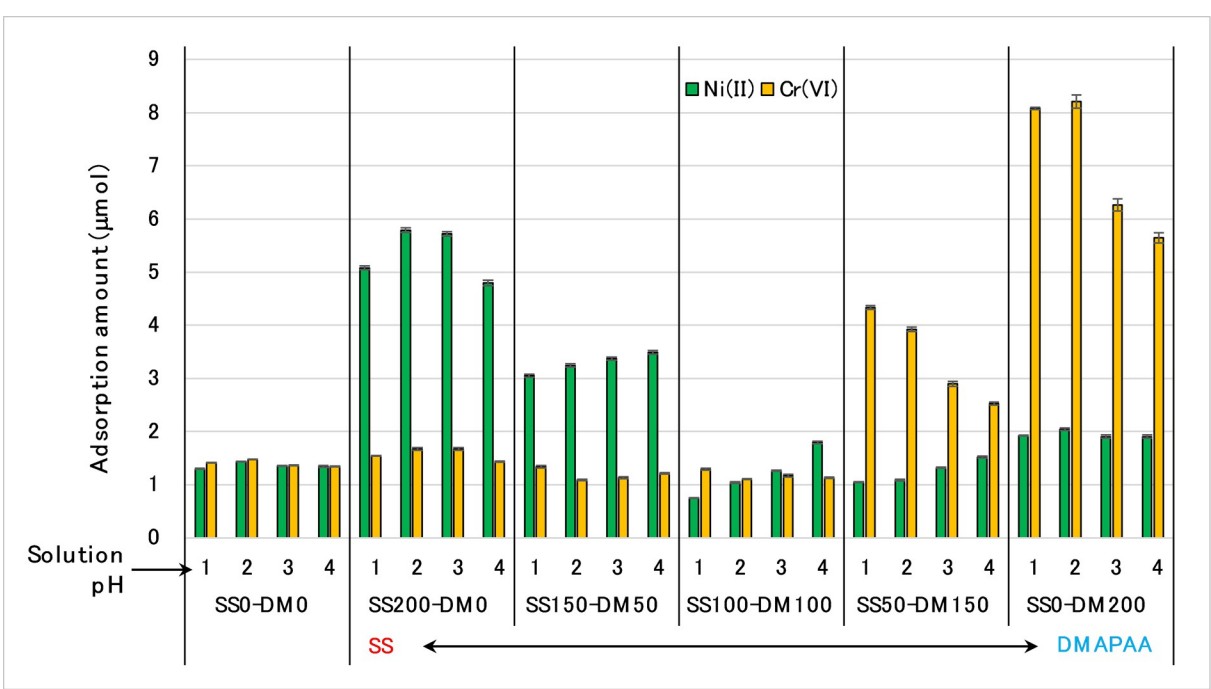

**Fig 5. Absorption profiles of (AAm–SS–DMAPAA) terpolymer hydrogel on Ni(II) and Cr(VI) from different pH conditions.**

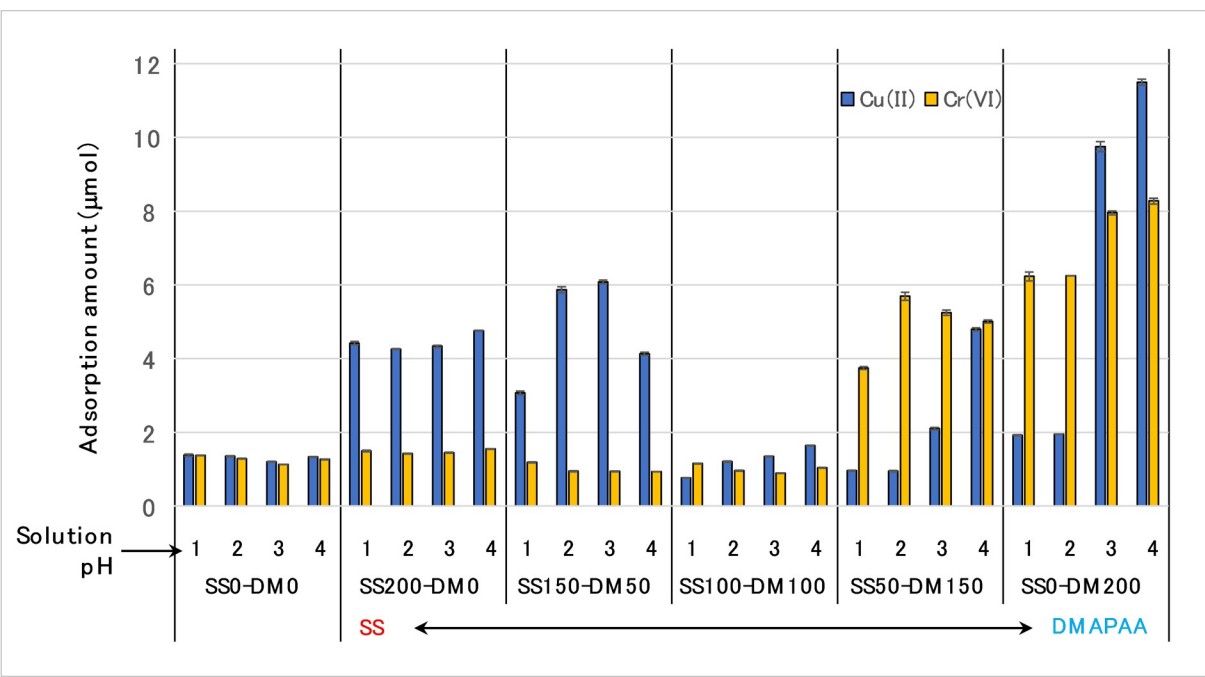

**Fig 6. Terpolymer hydrogel absorption behavior of Cu(II) and Cr(VI) from different pH conditions.**

DMAPAA (SS150 –DM 50) gel, the amount of $Cu^{2+}$ absorption was much higher than similar performance when using $Ni^{2+}$ as cation. This high $Cu^{2+}$ absorption was more prominent at pH 2 and 3 unlike in $Ni^{2+}/Cr^{6+}$ solution. When the amount of SS was maximum (i.e., SS200 –DM0 gel, which has no DMAPAA in the hydrogel network), it was expected that absorption of $Cu^{2+}$ would peak, but it was instead lower than in (SS150–DM50) gel.

More curiously, whereas when the DMAPAA concentration was gradually increased relative to SS, (SS50-DM150) gel, adsorptive removal of Cr(VI) was expectedly higher due to the effect of the DMAPAA. However, the uptake of $Cu^{2+}$ also significantly increased between pH 1–4 such that at pH 4, the (SS50-DM150) hydrogel absorption capacity of Cu(II) was nearly equal to Cr(VI) removal. When using solely the DMAPAA/AAm hydrogel (i.e., SS0–DM 200) gel, the phenomenon was even more pronounced. We may expect maximum anion capture and minimal cation capture, similar to what is shown in Fig 5. Instead, Fig 6 showed that the amount of Cr(VI) removal when using (SS0–DM 200) hydrogel was higher (8.2 µmol) but occurred at pH 3–4, which is contrary to this gel performance against $Ni^{2+}/Cr^{6+}$ solution (Fig 5) whose peak instead occurred at pH 1–2. Additionally, absorption of $Cu^{2+} >> Cr^{6+}$ which is contrarily to the expected $Cr^{6+} >> Cu^{2+}$ since higher DMAPAA amounts and no SS group in the hydrogel should favor absorption of only Cr(VI). It appears that due to the influence of DMAPAA, the poly(AAm-co-DMAPAA) hydrogel can be effectively utilized for high capacity $Cu^{2+}$ and $Cr^{6+}$ absorption.

Two hypotheses are proposed for this undue increase in $Cu^{2+}$ absorption especially at pH 3 and 4: either due to hydrolysis of the AAm main chain, or due to formation of metal hydroxides in the hydrogel[22.5].

*Hydrolysis of acrylamide (hypothesis 1).* Under basic conditions, AAm in the main polymer chain hydrolyzes to produce acrylic acid with ammonia side product according to the following Eq 3.

$$CH_2 = CHCONH_2 + H_2O \rightleftarrows CH_2 = CHCOOH + NH_3 \qquad (3)$$

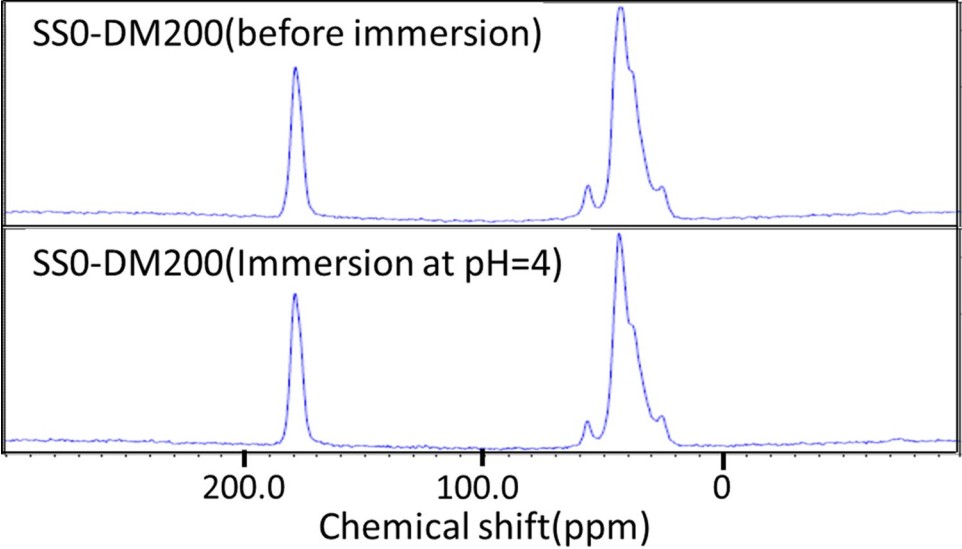

**Fig 7. NMR spectra of SS0-DM200 hydrogel before and after absorption of Cu(II) and Cr(VI).**

Therefore, considering the results in Fig 6, the authors postulated that at higher pH 3 and 4, the acrylate functional groups were responsible for increased absorption capacity of $Cu^{2+}$ via ion exchange in (SS0–DM 200) hydrogel by providing the necessary ion exchange sites. To confirm this, carbon-13 NMR was used to analyze the structure of the hydrogel after $Cu^{2+}$ ion absorption to find out whether a new peak of the carboxyl group was generated. However, the NMR chemical shifts in Fig 7 revealed no change in the peak waveform, therefore no structural changes in the hydrogel such as hydrolysis, occurred.

*Formation of metal hydroxides in hydrogels (hypothesis 2)*. Alternatively, the authors opined that it was likely metal hydroxides were generated in the solutions which had higher pH, during the absorption process. In SS0-DM200 hydrogel, the amine group of the DMAPAA could protonate in water to produce hydroxide ions as shown in the following Eq 4.

$$CH_2CHCONH(CH_2)_3N(CH_3)_2 + H_2O \leftrightarrow CH_2CHCONH(CH_2)_3N^+(CH_3)_2 + OH^- \quad (4)$$

The hydroxide ions generated would then be retained in a balanced ionic state, keeping the inside of the hydrogel basic. When this hydrogel was immersed in an aqueous cation-containing solution, the cations that diffused into the hydrogel reacted with the hydroxide ions to form metal hydroxides, which were probably collected in the hydrogel. Copper hydroxides have a significantly low theoretical solubility in water compared to Ni, at basic pH depending on concentrations in solution. The solubility product $K_{sp}$ of copper hydroxide and nickel hydroxide at 25°C is shown in Table 3.

Due to this difference in solubility product, the high absorption capacity of $Cu^{2+}$ than $Ni^{2+}$ could be attributed to precipitation of the $Cu^{2+}$, rather than ion exchange absorption.

**Table 3. Solubility product of copper hydroxide and nickel hydroxide at 25°C.**

| hydroxide | solubility product $K_{sp}$ $(mol/L)^2$ |
|---|---|
| $Cu(OH)_2$ | $1.58 \times 10^{-20}$ |
| $Ni(OH)_2$ | $5.01 \times 10^{-16}$ |

**Table 4. Swelling behavior of terpolymer hydrogel in different solution conditions.**

| Hydrogel type | | Swelling factor | | |
| --- | --- | --- | --- | --- |
| | | pure water | Ni(II), Cr(VI) | Cu(II), Cr(VI) |
| SS0-DM0 | pH = 1 | 1.4 | 1.0 | 1.2 |
| | pH = 2 | 1.5 | 1.1 | 1.2 |
| | pH = 3 | 1.5 | 1.1 | 1.1 |
| | pH = 4 | 1.5 | 1.1 | 1.1 |
| SS200-DM0 | pH = 1 | 11.7 | 1.8 | 1.9 |
| | pH = 2 | 11.8 | 2.0 | 1.9 |
| | pH = 3 | 12.3 | 2.1 | 2.0 |
| | pH = 4 | 12.4 | 1.8 | 2.1 |
| SS150-DM50 | pH = 1 | 7.6 | 1.0 | 1.1 |
| | pH = 2 | 7.4 | 1.2 | 1.3 |
| | pH = 3 | 7.4 | 1.2 | 1.3 |
| | pH = 4 | 7.4 | 1.2 | 1.2 |
| SS100-DM100 | pH = 1 | 4.3 | 0.4 | 0.4 |
| | pH = 2 | 4.4 | 0.5 | 0.4 |
| | pH = 3 | 4.4 | 0.6 | 0.4 |
| | pH = 4 | 4.4 | 0.6 | 0.5 |
| SS50-DM150 | pH = 1 | 1.3 | 1.0 | 1.1 |
| | pH = 2 | 1.3 | 1.0 | 1.3 |
| | pH = 3 | 1.4 | 1.0 | 1.2 |
| | pH = 4 | 1.5 | 1.0 | 1.1 |
| SS0-DM200 | pH = 1 | 6.3 | 2.3 | 2.3 |
| | pH = 2 | 6.3 | 2.4 | 2.3 |
| | pH = 3 | 6.1 | 2.2 | 2.1 |
| | pH = 4 | 6.1 | 2.1 | 2.1 |

Total monomer composition (AAm+SS+DMAPAA) = 700 mM in all cases.

**Terpolymer hydrogel absorption swelling behavior.** Swelling phenomenon during absorption by DMAPAA-SS/AAm terpolymer hydrogel absorbent on both $Ni^{2+}/Cr^{6+}$ and $Cu^{2+}/Cr^{6+}$ heavy metal solutions was analyzed by comparing its swelling capacity in pure water, and swelling in dual-heavy metal solution combinations of different pH, as in Table 4.

In the results summarized in Table 4, for each of six classes of terpolymer hydrogel, there were no disparities on the degree of swelling across different pH environment, thus the pH of metal solution was non-influencing on the swelling capacity.

Under pure water conditions of different pH, the control sample (i.e., SS0 –DM0 hydrogel) demonstrated a swelling factor of only X1.5. Yet, as the amount of SS gradually increased (from 50, 100, 150 and 200) relative to DMAPAA, the swelling factor rose sharply (X4.3, X7.4 reaching a peak of X12). Conversely, as the amount of DMAPAA increases relative to SS (i.e., from 50, 100, 150), the degree of swelling gradually decreases until a low of X1.5 in SS50–DM150 gel. However, at maximum DMAPAA concentration where SS is absent (i.e., SS0–DM200 hydrogel), swelling factor abruptly rises to X6. Thus, it is possible to say that both SS and DMAPAA are sufficiently hydrophilic to instigate water imbibition and higher swelling factor in the gel when existing independent of each other. However, when both SS and DMA-PAA coexist in the gel, higher DMAPAA amounts have adverse effect on the swelling mechanism, causing drastic decreases in the amount of swelling. So far, terpolymer hydrogel swelling only focused in pure water environments.

When the hydrogel swelling was studied under heavy metal anion/cation solution environments, only (SS200–DM0 hydrogel) and (SS0–DM200 hydrogel) showed significant swelling of X2. In both cases, only one of the ionic co-monomers was present together with AAm backbone monomer. If the SS and DMAPAA concentration is equal in the hydrogel (SS100–DM100 gel), the swelling factor in the hydrogel was lowest, at only ~X0.5 the initial weight. However, if the ratio of one of the ionic comonomer is higher than the other one, (i.e., SS150–DM50 gel, or SS50–DM150 gel) these hydrogel types yielded a swelling factor of approximately X1. Overall, when this low swelling factors in heavy metal environments was compared with the high swelling under "ideal" pure water conditions, the hydrogel is considered to have "shrunk".

Swelling (in pure water) and shrinkage (in heavy metal water) is a unique phenomenon in hydrogel absorption mechanism that sets it apart from other type of absorbents, which do not imbibe water nor swell. In heavy metal anion/cation solutions, since the metal ions are already dissolved in water, therefore, water imbibition into the hydrogel acts as transport mechanism that facilitates transfer of the dissolved heavy metals from the solution into the interior channels of the hydrogel polymer network. As the hydrogel swells and increases in size, these metal ions are then able to access and interact with the previously inaccessible interior active sites of the absorbent, thereby ensuring maximum utilization of all active sites during absorption, ultimately ensuring higher absorption capacity. Therefore, in hydrogels, absorption occurs both on the surface (mainly), and on inner active sites of the hydrogel network. Secondly, when the metals are absorbed onto the polymer networks, this (metal + functional group) interaction causes inter-linking of participating polymer chains, creating some kind of secondary cross-links within the bulk of the material. This makes the post-absorption polymer networks stiffer. As more metals are absorbed, the chains become stiffer. Consequently, due to stiffness/secondary cross-links, the post-absorption state hydrogel cannot swell as much as in ideal conditions, and this is why there is low swelling factor (resembling shrinkage) in metal-absorbed state than normal swelling in pure distilled water. This phenomenon was illustrated in Fig 8.

## Copolymer hydrogel absorption, swelling and effect of internal pH environment

The authors also analyzed the effect of only DMAPAA on the absorption behavior of heavy metal ions, using combined solution comprising Cr(VI), Ni(II), Cu(II) and Zn(II) metal ions. Different sets of AAm-co-DMAPAA copolymer hydrogels were synthesized with increasing ratios of DMAPAA (see Table 2 earlier). A homopolymer hydrogel using only AAm was used as control sample. The effect of the internal pH environment of the freshly synthesized hydrogels was also investigated, by measuring the pH of each hydrogel prior to any application on absorption solutions. When these hydrogels were used for batch absorption of the multi-metal solution, the resultant absorption capacity was compared with the respective hydroxide concentration of each hydrogel's internal pH condition.

Table 5 highlights results on initial condition of the copolymer hydrogels including the internal pH, and swelling behavior in pure water and post-absorption swelling in heavy metal water environment. In pure water, DM100 hydrogel achieved a swelling factor of X3, and the swelling capacity gradually increased as the DMAPAA concentration gradually increased until a maximum swelling of X6.7, thus confirming the strongly hydrophilic nature of this ionic group. However, unlike the terpolymer hydrogel (in Table 4 previously) which did not swell after absorption of heavy metals, in present case the AAm-co-DMAPAA copolymer hydrogels was able to achieve increasingly higher swelling capacities even after absorption of metal ions, with up to 400% mass increase at maximum DMAPAA concentration. Therefore, if only one

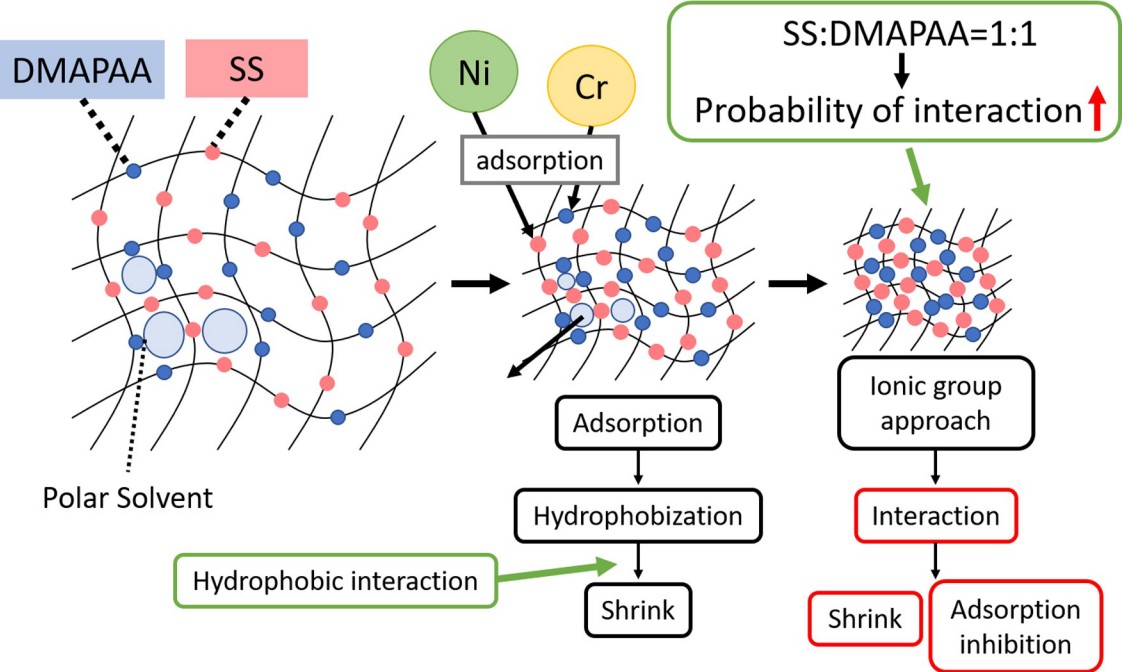

**Fig 8. Metal-uptake and swelling mechanism of AAm–SS–DMAPAA hydrogel from Ni/Cr solution.**

ionic monomer is present, the hydrogel is able to achieve significant swelling in heavy metal solution. If both SS and DMAPAA are present, two scenarios are possible: (1) the two groups reinforce each other to result in stronger electrostatic interaction with metal ions during ion exchange absorption. This strong interaction greatly constricts the polymer chains, impeding any ability to swell. (2) Alternatively, as postulated in Fig 8, the two functional groups may also be antagonistic to each other, creating hydrophobic interaction that expels water from the hydrogel network during absorption process. But when only one ionic monomer is present such as DMAPAA, then absorption swelling in heavy metal water is significantly higher, approx. 55% of the swelling factor in pure water conditions.

Further analysis of absorption results in Fig 9 revealed that indeed as the concentration of DMAPAA monomer gradually increased in the hydrogel, concurrently its absorption capacity of $Cr^{6+}$ increased, reaching a peak at DM600 hydrogel where DMAPAA concentration was

**Table 5. Intrinsic conditions and swelling properties of AAm-co-DMAPAA copolymer hydrogel.**

| Hydrogel type | Internal pH | Swelling in water | Swelling in metal solution |
|---|---|---|---|
| DM 0 | 2.80 | 1.5 | 1.0 |
| DM 100 | 8.84 | 3.1 | 1.5 |
| DM 200 | 9.14 | 3.2 | 2.0 |
| DM 300 | 9.56 | 4.6 | 2.6 |
| DM 400 | 9.64 | 5.7 | 3.1 |
| DM 500 | 9.82 | 6.4 | 3.6 |
| DM 600 | 9.89 | 6.7 | 4.0 |

Total monomers composition (AAM + DMAPAA) = 700 mM in all cases.

The initial concentration of metal solution ($Cr^{6+}$, $Ni^{2+}$, $Cu^{2+}$, $Zn^{2+}$) used = 5 mM.

highest. However, $Cu^{2+}$ absorption also increased in a similar fashion, but at much higher rate than $Cr^{6+}$ uptake by these hydrogels. Meanwhile, absorption of the other metal ions remained very low in all cases, i.e., absorption of $Cu^{2+} > Cr^{6+} >> Ni^{2+}, Zn^{2+}$. Additionally, the hydrogel's pH environment appeared to be strongly influencing on the degree of absorption, since as the hydroxide concentration increased, the hydrogel's absorption capacity of $Cr^{6+}$ and $Cu^{2+}$ correspondingly increased.

The internal pH of all copolymer hydrogel samples in Table 5 showed that whereas the control sample was acidic in nature, upon introduction of slight concentration of DMAPAA comonomer, the pH of the hydrogel internal environment immediately flips to basic, and the basicity rises with higher DMAPAA concentrations. It was possible to calculate the concentration of hydroxide ions in the hydrogel using the following Eq 5 (at 25˚C):

$$[OH^-] = 10^{(pH-14)} \tag{5}$$

Bo correlating the hydroxide concentration and absorption behavior (Fig 9), it was possible to infer that higher DMAPAA concentrations was responsible for higher hydroxide concentration in the hydrogels, and correspondingly higher absorption of Cu(II) cations and Cr(VI) anions. However, compared to other cations, only the absorption of Cu(II) cations exceeded Cr(VI) capture by the DMAPAA functional group. Therefore, it was highly probable that

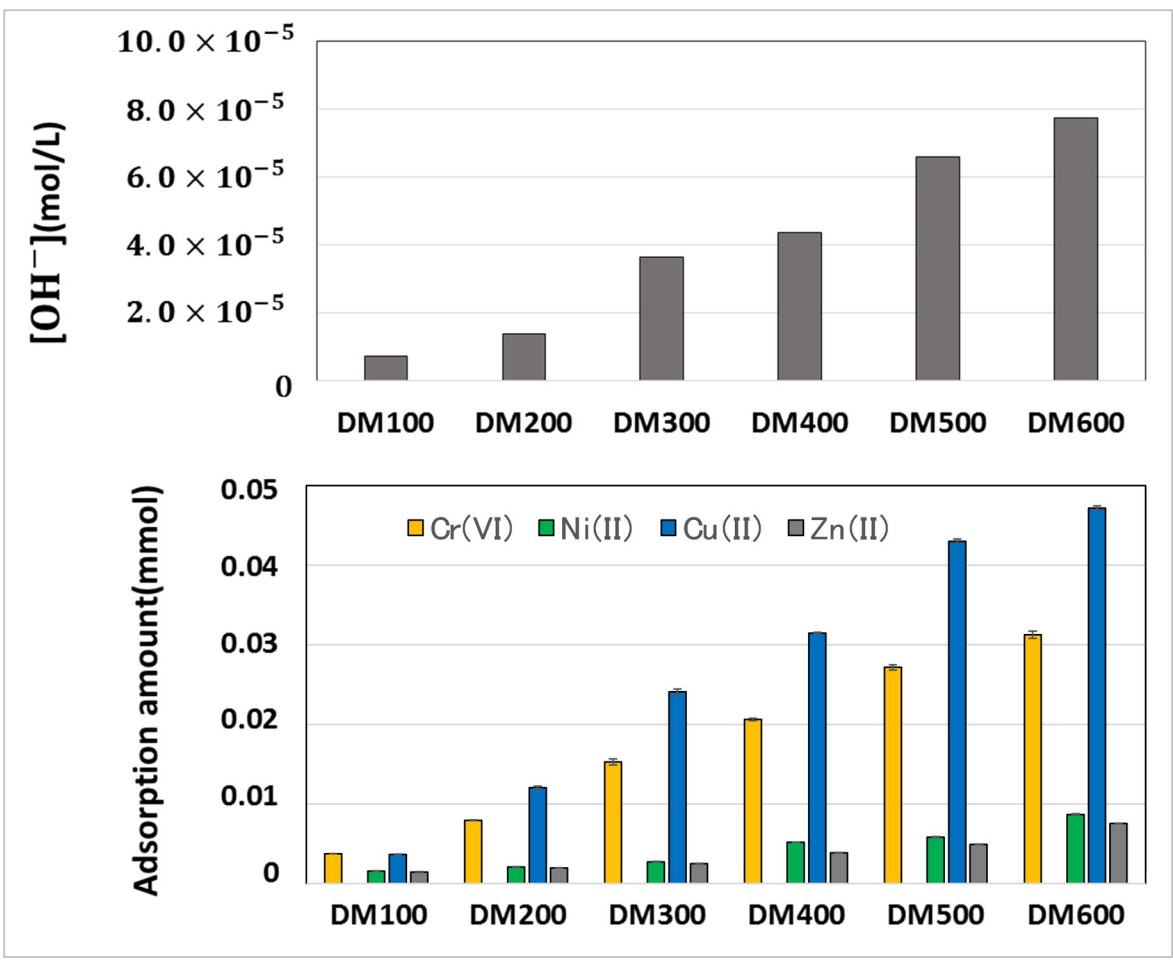

**Fig 9. Copolymer hydrogels internal pH condition versus respective metal absorption capacities.**

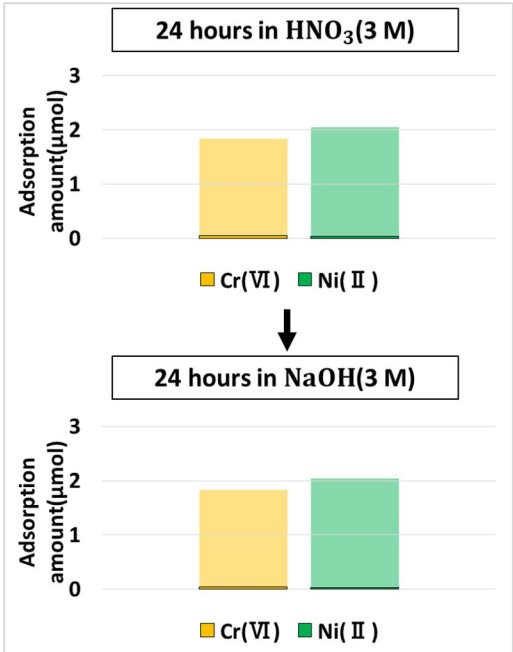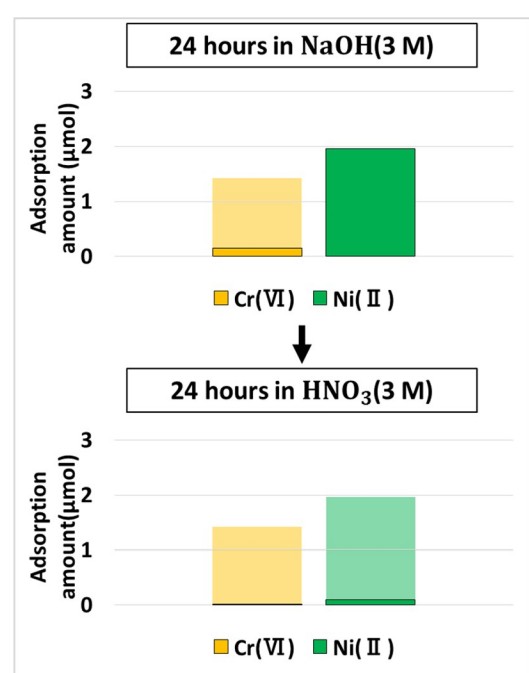

**Fig 10. Desorption protocols of Cr(VI) and Ni(II) embedded into (AAm–SS–DMAPAA) hydrogel.**

unlike $Ni^{2+}$ and $Zn^{2+}$ which experienced mainly due to action of AAm functional group (low absorption capacity), the $Cu^{2+}$ was adversely affected by the high pH environment of the hydrogel due to effect of high DMAPAA concentration, and consequently the copper ions were precipitated from the solution. Higher DMAPAA ratios increases the basicity inside the hydrogel and since copper has a lower solubility threshold (equivalence point) at a given concentration compared to other metals like Ni(II) and Zn(II). Therefore, the high basicity inside the hydrogel easily caused them to precipitate as copper hydroxide, hence giving a false signal of absorption when the final solution concentration was analyzed.

This proposal was confirmed by analyzing the physical properties of hydrogel samples prior to and after absorption of heavy metals as pictured in S5 Fig. Original samples are clear and consistent, but after metal absorption of each hydrogel from the four metal solutions, S5 Fig shows that the turbidity of the final hydrogel gradually increases with higher DMAPAA concentration in the hydrogel. This was attributed to fine particles of metal hydroxide (copper hydroxide) with a size larger than the wavelength of light that are deposited inside the hydrogel due to metal precipitation, contributing to the turbid effect. Based on these conclusions, the DMAPAA hydrogels can be described as amphoteric gels, due to its ability to remove both cationic ($Cu^{2+}$) and anionic ($Cr^{6+}$) heavy metals with high efficiency, from concentrated solutions.

## Desorption experiments on DMAPAA hydrogel

Desorption experiments were performed by immersing the metal-absorbed hydrogels in concentrated acid and alkaline solutions in reverse orders. In the first case, using the SS100–D100 hydrogel that was used for the absorption of Ni(II) and Cr(VI) capture, the hydrogel was initially desorbed via immersion in concentrated $HNO_3$ followed by simultaneous immersion into 3M NaOH solution. The order of immersion was then reversed, in order to find out selectivity in metal elution, and effectiveness in metal desorption and absorbent regeneration. The

results are summarized in Fig 10. In all cases, faint colors will refer to the total amount absorbed after simultaneous absorption, whereas the lower dark bar reflects the metal amount retained in the hydrogel even after desorption process.

Obtained results showed that in both cases, all the metals were eluted from the absorbent, but the process of immersion in $HNO_3$ followed by NaOH was not suitable for selective absorption of cations and anions since both metals were summarily desorbed. But in the reverse process, only Cr(VI) anion was selectively desorbed by the NaOH, and when the absorbent was later immersed into $HNO_3$, the remaining Ni(II) within the hydrogel was finally eluted by the concentrated acid. Indicating that to achieve selectivity in the order of metal elution from absorbent network, it is imperative to follow the order of immersion into NaOH followed by $HNO_3$.

Accordingly, for the hydrogel samples that were used for batch absorption of a mixture solution of Ni(II), Cu(II), Zn(II), and Cr(VI) for 72 hours, desorption process followed the order of NaOH and $HNO_3$ immersion. Desorption process was also modified by reducing the size of the hydrogel (from 0.25 g to 0.1 g segments) in order to increase the surface area of interaction with the eluent solution. Secondly, the range of acid and alkaline solution was varied from 0.3 M to 3 M to find out corresponding influence on degree of metal elution. In the main finding depicted in Fig 11, faint colors will similarly refer to the total absorbed amount, whereas the lower dark bar reflects the metal amount still retained in the hydrogel even after desorption process. The results indicated that immersion into only alkaline (NaOH) solution is insufficient to elute the metals from the absorbent. Secondly, lower concentrations of $HNO_3$ than 3M causes incomplete desorption with large amounts of metals still retained within the hydrogel. Also, despite low concentrations of NaOH in the first cycle, if the $HNO_3$ is maintained at 3M in the second cycle, total desorption of all metals would still be achieved.

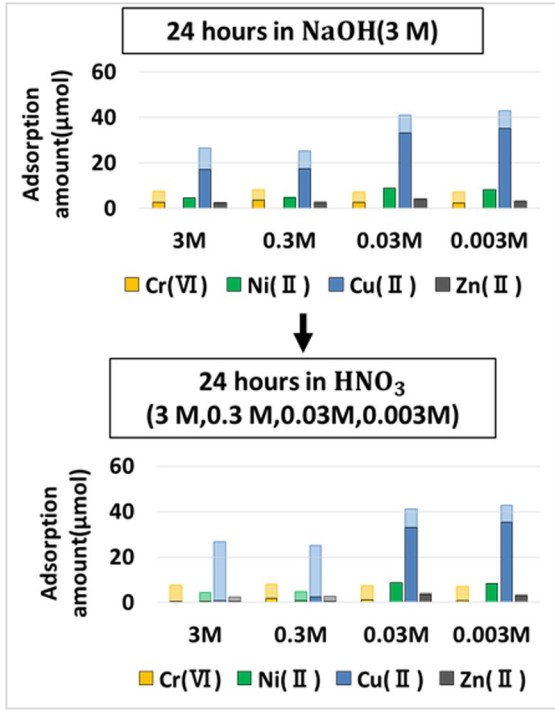
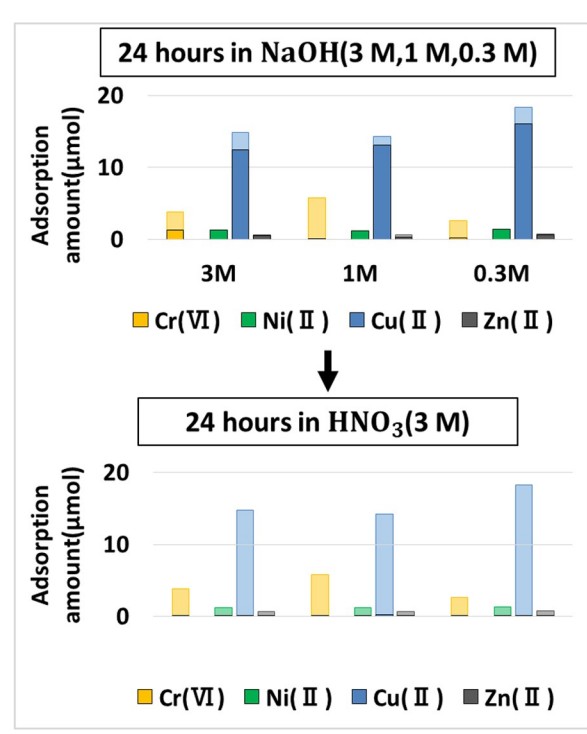

**Fig 11. Desorption protocols of mixed metals from the AAm–co-DMAPAA hydrogels.**

Pre-treatment of the hydrogel absorbents in NaOH solutions causes the metals embedded in the hydrogel to be sufficiently mobile, such that when this hydrogel was later washed in concentrated nitric acid, all the metals are completely desorbed from the hydrogel network and transferred into the nitric acid solution. The dark bar (which indicates the amount of metal retained in the hydrogel) completely disappeared, proving high degree of desorption. Additionally, all the cations were uniformly desorbed by 3 M nitric acid, indicating the separation and desorption of cations and anions.

## Conclusion

In this research, we synthesized various AAm hydrogels incorporating SS and DMAPAA co-monomers, aiming to create ionized polymer hydrogels capable of separating and recovering heavy metals from effluents with mixed cations and anions. Our study focused on the hydrogels' ability to absorb and desorb heavy metal ions from concentrated solutions, yielding several key findings:

1. The use of DMAPAA hydrogel enabled the concurrent absorption and subsequent separation/desorption of heavy metal cations and anions. The SS co-monomer was effective in capturing cations, while DMAPAA facilitated the extraction of anionic species such as Cr(VI).

2. We observed that the absorption capacity was influenced by the concentration of SS or DMAPAA in the terpolymer hydrogel. It is crucial to maintain a higher concentration of either SS or DMAPAA, depending on the intended application, to ensure optimal efficiency.

3. In the copolymer hydrogels (comprising only AAm and DMAPAA), a higher proportion of DMAPAA not only absorbed Cr(VI) anions but also showed significant extraction of copper ions. This higher extraction of Cr6+ can be attributed to the electrostatic attraction between DMAPAA and chromium anions. In contrast, copper ion removal was due to the creation of a highly basic environment within the hydrogel network by DMAPAA, leading to the precipitation and deposition of copper ions. This highlights the amphoteric nature of the DMAPAA monomer, capable of efficiently absorbing both cationic (Cu2+) and anionic (Cr6+) species.

4. Both DMAPAA and SS are hydrophilic ionic monomers, which contribute to the high swelling capacities of the bipolar hydrogel. By examining these monomers individually and in combination within copolymer and terpolymer hydrogels, we were able to assess the extent of inhibition caused by both polarities. The presence of both monomers in a gel results in reduced absorption capacities due to DMAPAA counteracting SS's hydrophilicity. In the context of heavy metal solutions, the swelling capacity of the post-absorption AAm-SS-DMAPAA terpolymer hydrogels is significantly limited, primarily due to the formation of secondary cross-links during metal absorption and the strong interaction between metal ions and the functional groups of the monomers, which enhances their combined metal-capturing efficiency.

5. For effective desorption and separation of metal cations (such as Cu, Ni, Zn) and anionic groups (e.g., Cr(VI)), the metal-absorbed hydrogels should be segmented into smaller pieces and treated in concentrated solutions sequentially with sodium hydroxide followed by 3M nitric acid, each for 24-hour periods.

In conclusion, while the current gel composition holds potential for the simultaneous recovery of both anionic and cationic heavy metal species from the waste liquids in the metal

plating industry, efficiency improvements are necessary. Specifically, reducing the interaction between the anion and cation adsorption groups is critical. Utilizing a two-step process with a copolymer that selectively adsorbs either anions or cations could undermine the benefits of the terpolymer. Therefore, future strategies should focus on phase-separating SS and DMAPAA during the synthesis of block terpolymers. Successful implementation of this approach will pave the way for field testing.

## Supporting information

**S1 Fig. The FT-IR spectra of 4 samples of different SS and DMAPAA compositions.** In the spectrum obtained from the gel prepared using SS, a peak corresponding to S = O stretching at a wave number of 1200–1185 cm$^{-1}$ derived from SS is observed. This suggests that SS was introduced. On the other hand, a change in the amide band II at a wave number of 1550–1590 cm$^{-1}$ derived from DMAPAA can be observed in the gel using DMAPAA, but this change is also due to AAm, and it also appears in the gel without DMAPAA. So, it is not possible to judge the introduction of DMAPAA. FT-IR measurements were performed using the gel samples that were dried at 60˚C under pressure to make them flat. All data were measured using FT/IR-4000 with Attenuated Total Reflection (ATR) attachment (JASCO corp., Japan). (TIF)

**S2 Fig. Powder samples for NMR measurements.** After thoroughly drying the gels, they were ground into powders in a mortar. (TIF)

**S3 Fig. NMR original data before being summarized in Fig 4 and other data measured at the same time.** (a) $^1$H and $^{13}$C spectra of SS0-DM0 sample (i.e., AAm gel). (b) $^1$H and $^{13}$C spectra of SS200-DM0 copolymer gel. (c) $^1$H and $^{13}$C spectra of SS0-DM200 copolymer gel. (d) $^1$H and $^{13}$C spectra of SS100-DM100 terpolymer gel. The measuring conditions are shown in their right column. The $^1$H peaks don't provide any useful structural information, but the $^{13}$C peaks are unique to each monomer. Please refer main part. (TIF)

**S4 Fig. NMR original data before being summarized in Fig 7 and other data measured at the same time.** (a) $^1$H and $^{13}$C spectra of SS200-DM0 sample. Data obtained by re-measuring the sample with the same components as S3(A) Fig. (b) $^1$H and $^{13}$C spectra of SS0-DM0 sample (i.e., AAm gel) immersed in pH4 HNO$_3$ solution for 72 hours. (c) $^1$H and $^{13}$C spectra of SS0-DM200 copolymer gel. Data obtained by re-measuring the sample with the same components as S3(C) Fig. (d) $^1$H and $^{13}$C spectra of SS100-DM100 terpolymer gel immersed in pH4 HNO$_3$ solution with Cu$^{2+}$ and (Cr$_2$O$_7$)$^{2-}$ for 72 hours. (e) $^1$H and $^{13}$C spectra of SS100-DM100 terpolymer gel immersed in pH4 HNO$_3$ solution for 72 hours. (f) $^1$H and $^{13}$C spectra of SS100-DM100 terpolymer gel immersed in pH1 HNO$_3$ solution for 72 hours. The measuring conditions are shown in their right column. The $^{13}$C spectra in figures (c) and (d) are used to make Fig 7. (TIF)

**S5 Fig.** Pre-absorption (a) and post-absorption (b) profiles of AAm–DMAPAA hydrogels. Both photos show DM100, DM200, DM300, DM400, DM500, and DM600 from the left, and the higher the DMAPAA concentration, the more turbidity occurs after adsorption. It is thought that fine particles of copper hydroxide are trapped inside the gel. (TIF)

## Acknowledgments

The NMR measurements were supported by Evaluation Center of Materials Properties and Function, Institute for Materials Chemistry and Engineering, Kyushu University.

## Author Contributions

**Conceptualization:** Kazuhiro Hara.

**Data curation:** Kentaro Fujimoto.

**Funding acquisition:** Hirotaka Okabe, Kazuhiro Hara.

**Investigation:** Kentaro Fujimoto, Brian Adala Omondi, Shinya Kawano, Yoshiki Hidaka, Kenji Ishida, Hirotaka Okabe, Kazuhiro Hara.

**Methodology:** Kazuhiro Hara.

**Project administration:** Hirotaka Okabe, Kazuhiro Hara.

**Supervision:** Hirotaka Okabe, Kazuhiro Hara.

**Validation:** Brian Adala Omondi, Hirotaka Okabe, Kazuhiro Hara.

**Visualization:** Kentaro Fujimoto.

**Writing – original draft:** Kentaro Fujimoto.

**Writing – review & editing:** Brian Adala Omondi, Hirotaka Okabe.

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
