## [Decision Letter · Decision Letter 0]

10 Oct 2023

PONE-D-23-29103Ionized acrylamide copolymer and terpolymer hydrogel absorbents for fractional recovery of positive and negative heavy metal ions from wastewaterPLOS ONE

Dear Dr. Okabe,

Thank you for submitting your manuscript to PLOS ONE. After careful consideration, we feel that it has merit but does not fully meet PLOS ONE’s publication criteria as it currently stands. Therefore, we invite you to submit a revised version of the manuscript that addresses the points raised during the review process.

We look forward to receiving your revised manuscript.

Kind regards,

Nayan Ranjan Singha, Ph.D.

Academic Editor

PLOS ONE

Journal Requirements:

2. We note that this submission includes NMR spectroscopy data. We would recommend that you include the following information in your methods section or as Supporting Information files:

a) The make/source of the NMR instrument used in your study, as well as the magnetic field strength. For each individual experiment, please also list: the nucleus being measured; the sample concentration; the solvent in which the sample is dissolved and if solvent signal suppression was used; the reference standard and the temperature.

b) A list of the chemical shifts for all compounds characterised by NMR spectroscopy, specifying, where relevant: the chemical shift (δ), the multiplicity and the coupling constants (in Hz), for the appropriate nuclei used for assignment.

c)The full integrated NMR spectrum, clearly labelled with the compound name and chemical structure.

We also strongly encourage authors to provide primary NMR data files, in particular for new compounds which have not been characterised in the existing literature. Authors should provide the acquisition data, FID files and processing parameters for each experiment, clearly labelled with the compound name and identifier, as well as a structure file for each provided dataset. See our list of recommended repositories here: https://journals.plos.org/plosone/s/recommended-repositories

   "This work was supported by JSPS KAKENHI Grant Numbers JP21656239, JP24360398, JP19H02660, JP21H01871. The NMR measurements were supported by Evaluation Center of Materials Properties and Function, Institute for Materials Chemistry and Engineering, Kyushu University."

  "This work was supported by JSPS KAKENHI Grant Numbers JP21656239, JP24360398, JP19H02660, JP21H01871. "

   "This work was supported by JSPS KAKENHI Grant Numbers JP21656239, JP24360398, JP19H02660, JP21H01871. "

Additional Editor Comments:

Comments from the Editor

1. The authors should provide at the first revision stage: Details of the NMR instrument and method used, a full list of chemical shifts, and labelled integrated copies of the NMR spectra.

2. Title should be changed to the following:

Ionized acrylamide-based copolymer/ terpolymer hydrogels for recovery of positive and negative heavy metal ions

3. Both abstract and conclusion should be more focused.

Reviewers' comments:

Reviewer's Responses to Questions

**Comments to the Author**

1. Is the manuscript technically sound, and do the data support the conclusions?

Reviewer #1: Partly

Reviewer #2: Partly

2. Has the statistical analysis been performed appropriately and rigorously? 

Reviewer #1: No

Reviewer #2: N/A

3. Have the authors made all data underlying the findings in their manuscript fully available?

Reviewer #1: Yes

Reviewer #2: No

4. Is the manuscript presented in an intelligible fashion and written in standard English?

Reviewer #1: Yes

Reviewer #2: Yes

5. Review Comments to the Author

Reviewer #1: Manuscript entitled “Ionized acrylamide copolymer and terpolymer hydrogel absorbents for fractional recovery of positive and negative heavy metal ions from wastewater” submitted by Kentaro Fujimoto, Brian Adala Omondi, Shinya Kawano, Yoshiki Hidaka, Kenji Ishida, Hirotaka Okabe, Kazuhiro Hara, can be considered for publication in PLOS ONE Journal, after a serious major revisions.

Here is a list of my specific comments:

1. General comment: The novelty and practical applicability of this study should be clearly highlighted in the manuscript.

2. Page 1, Abstract: (a) Delete the first sentence. (b) Add in this section the most important experimental results to highlight the importance of this study. (c) What are the ions selected for this study???

3. Page 1, Introduction: This section should be reorganized. The most important aspects related to this topic must be clearly presented to provide a properly presentation of the state of art in this field. Also, the metal ions selected for this study must be mentioned.

4. Page 2, line 40: “Methodologies for heavy metal wastewater…”. Add here as reference the papers: doi.org/10.1016/j.jclepro.2018.06.261 and doi.org/10.3390/polym15102301, because are relevant for this observation.

5. Page 2, line 49: “Cr(VI) is a notorious pollutant due to its…”. Add here some references.

6. Page 3, line 75: “…selected heavy metal cations,…”. The metal ions selected for this study should be mentioned.

7. Page 3, line 79: “for reuse[18]–[20].”. Delete these references, because here are presented the objectives of this study.

8. Page 3, line 81: The same observation as above.

9. Page 5, line 130: Replace “ICP/MS machine” with “ICP/MS equipment (or spectrometer)”.

10. Page 5, The effect of DMAPAA in (AAm – DMAPAA) copolymer hydrogels metal absorption: Some paragraphs have be already mentioned before. These should be deleted. Also, provide a clear presentation of the experimental methodology.

11. Page 7, Structural analysis using NMR: These experimental results must be more detailed discussed.

12. Page 7, Absorption from combined heavy metal cation/anion solutions: This section should be reorganized. All experimental results must be clearly presented and detailed discussed. Irrelevant general observations should be deleted.

13. Page 10, Terpolymer hydrogel absorption swelling behavior: The same observations as above.

14. Page 13, Copolymer hydrogel absorption, swelling and effect of internal pH environment: The same observations.

15. Page 17, Conclusion: This section must be reorganized. Delete (1), (2),…, and provide in this section a clear presentation of the most important experimental results and findings.

16. Page 18, References: The number of references must be increased.

17. Fig. 8 should be moved into Supplementary materials.

Reviewer #2: Dear Authors

Thank you for your interest in the PLOS ONE. I have received your manuscript (PONE-D-23-29103) entitled “Ionized acrylamide copolymer and terpolymer hydrogel absorbents for fractional recovery of positive and negative heavy metal ions from wastewater". However, after reading your paper, you should take the comments very carefully. My opinion is that the manuscript should be Reviewed again after Major revision and the manuscript should be resubmitted.

The following are comment to the author:-

1- Chemical structure of the used materials should be provided in experimental part.

2- FTIR analysis should be performed for some selected prepared terpolymers and also for mono-polymers.

3- All analytical devices should be mentioned separately in the experimental part with its manufacturing countries, model, and condition for each tool.

4- In results and discussion there is no equation for the protonation of (SS) like that of AAm and (DMAPAA).

5- References should be added for the protonation and dissociation of both AAm and (DMAPAA).

6- All equations should be numbered.

7- NMR not discussed in the manuscript, the authors should add brief discussion.

8- Figure captions should be added at the end of each provided figure.

9- In electroplating industries as mentioned in introduction part, where the authors can apply their best hydrogel composition, how they can change the pH of the resulting actual wastewater?

6. PLOS authors have the option to publish the peer review history of their article (what does this mean?). If published, this will include your full peer review and any attached files.

Reviewer #1: No

Reviewer #2: **Yes: **Prof. A. M. Abdel-Ghaffar

---

## [Author Response · Author response to Decision Letter 0]

12 Jan 2024

Dear Editor,

The author who originally wrote the manuscript and the author who took over this have moved on for work, and I need time to communicate with them, so the revisions have been delayed, but we appreciate your continued support. We thought it would be easier to understand if we wrote them in the email text we received, so we wrote our response after mark of ****. We have replied each item below.

Best regards,

Hirotaka Okabe and other authors

PONE-D-23-29103

Ionized acrylamide copolymer and terpolymer hydrogel absorbents for fractional recovery of positive and negative heavy metal ions from wastewater

PLOS ONE

Dear Dr. Okabe,

Thank you for submitting your manuscript to PLOS ONE. After careful consideration, we feel that it has merit but does not fully meet PLOS ONE’s publication criteria as it currently stands. Therefore, we invite you to submit a revised version of the manuscript that addresses the points raised during the review process.

We look forward to receiving your revised manuscript.

Kind regards,

Nayan Ranjan Singha, Ph.D.

Academic Editor

PLOS ONE

Journal Requirements:

**** We intend to follow all instructions.

2. We note that this submission includes NMR spectroscopy data. We would recommend that you include the following information in your methods section or as Supporting Information files:

a) The make/source of the NMR instrument used in your study, as well as the magnetic field strength. For each individual experiment, please also list: the nucleus being measured; the sample concentration; the solvent in which the sample is dissolved and if solvent signal suppression was used; the reference standard and the temperature.

b) A list of the chemical shifts for all compounds characterised by NMR spectroscopy, specifying, where relevant: the chemical shift (δ), the multiplicity and the coupling constants (in Hz), for the appropriate nuclei used for assignment.

c)The full integrated NMR spectrum, clearly labelled with the compound name and chemical structure.

We also strongly encourage authors to provide primary NMR data files, in particular for new compounds which have not been characterised in the existing literature. Authors should provide the acquisition data, FID files and processing parameters for each experiment, clearly labelled with the compound name and identifier, as well as a structure file for each provided dataset. See our list of recommended repositories here: https://journals.plos.org/plosone/s/recommended-repositories

**** We provide all original NMR data with the measuring condition in Supporting Information.

 "This work was supported by JSPS KAKENHI Grant Numbers JP21656239, JP24360398, JP19H02660, JP21H01871. The NMR measurements were supported by Evaluation Center of Materials Properties and Function, Institute for Materials Chemistry and Engineering, Kyushu University."

 "This work was supported by JSPS KAKENHI Grant Numbers JP21656239, JP24360398, JP19H02660, JP21H01871. "

**** We would like to remove "This work was supported by JSPS KAKENHI Grant Numbers JP21656239, JP24360398, JP19H02660, JP21H01871. The NMR measurements were supported by Evaluation Center of Materials Properties and Function, Institute for Materials Chemistry and Engineering, Kyushu University." from the manuscript, and add “The NMR measurements were supported by Evaluation Center of Materials Properties and Function, Institute for Materials Chemistry and Engineering, Kyushu University." to Funding Statement.

**** As same as the assignment of 3.

 "This work was supported by JSPS KAKENHI Grant Numbers JP21656239, JP24360398, JP19H02660, JP21H01871. "

**** Please add "The funders had no role in study design, data collection and analysis, decision to publish, or preparation of the manuscript.".

Additional Editor Comments:

Comments from the Editor

1. The authors should provide at the first revision stage: Details of the NMR instrument and method used, a full list of chemical shifts, and labelled integrated copies of the NMR spectra.

**** We provide all original NMR data with the measuring condition in Supporting Information.

2. Title should be changed to the following:

Ionized acrylamide-based copolymer/ terpolymer hydrogels for recovery of positive and negative heavy metal ions

**** We change the title as suggested.

3. Both abstract and conclusion should be more focused.

**** We intend to rewrite it as instructed.

Reviewers' comments:

Reviewer's Responses to Questions

Comments to the Author

1. Is the manuscript technically sound, and do the data support the conclusions?

Reviewer #1: Partly

Reviewer #2: Partly

2. Has the statistical analysis been performed appropriately and rigorously? 

Reviewer #1: No

Reviewer #2: N/A

3. Have the authors made all data underlying the findings in their manuscript fully available?

Reviewer #1: Yes

Reviewer #2: No

4. Is the manuscript presented in an intelligible fashion and written in standard English?

Reviewer #1: Yes

Reviewer #2: Yes

Dear Reviewer 1,

The author who originally wrote the manuscript and the author who took over this have moved on for work, and I need time to communicate with them, so the revisions have been delayed, but I appreciate your continued support. Thank you for your valuable feedback. In consideration of your comments, we have made as many corrections as possible. We thought it would be easier to understand if we wrote them in the email text we received, so we wrote our response after mark of ****. We have replied each item below.

Best regards,

Hirotaka Okabe and other authors

5. Review Comments to the Author

Reviewer #1: Manuscript entitled “Ionized acrylamide copolymer and terpolymer hydrogel absorbents for fractional recovery of positive and negative heavy metal ions from wastewater” submitted by Kentaro Fujimoto, Brian Adala Omondi, Shinya Kawano, Yoshiki Hidaka, Kenji Ishida, Hirotaka Okabe, Kazuhiro Hara, can be considered for publication in PLOS ONE Journal, after a serious major revisions.

Here is a list of my specific comments:

1. General comment: The novelty and practical applicability of this study should be clearly highlighted in the manuscript.

**** We intend to rewrite it as you said.

2. Page 1, Abstract: (a) Delete the first sentence. (b) Add in this section the most important experimental results to highlight the importance of this study. (c) What are the ions selected for this study???.

**** We rewrote the Abstract as suggested.

3. Page 1, Introduction: This section should be reorganized. The most important aspects related to this topic must be clearly presented to provide a properly presentation of the state of art in this field. Also, the metal ions selected for this study must be mentioned.

**** We rewrote the Abstract as suggested.

4. Page 2, line 40: “Methodologies for heavy metal wastewater…”. Add here as reference the papers: doi.org/10.1016/j.jclepro.2018.06.261 and doi.org/10.3390/polym15102301, because are relevant for this observation.

**** We added the references.

5. Page 2, line 49: “Cr(VI) is a notorious pollutant due to its…”. Add here some references.

**** We added the reference.

6. Page 3, line 75: “…selected heavy metal cations,…”. The metal ions selected for this study should be mentioned.

**** The ionic formulas were shown.

7. Page 3, line 79: “for reuse[18]–[20].”. Delete these references, because here are presented the objectives of this study.

**** We think your comments are useful, but we think it would be better to leave it as a reference for many readers.

8. Page 3, line 81: The same observation as above.

**** We repeated for emphasis. Is it better to summarize them?

9. Page 5, line 130: Replace “ICP/MS machine” with “ICP/MS equipment (or spectrometer)”.

**** We fixed.

10. Page 5, The effect of DMAPAA in (AAm – DMAPAA) copolymer hydrogels metal absorption: Some paragraphs have be already mentioned before. These should be deleted. Also, provide a clear presentation of the experimental methodology.

**** We think your points are useful, but many researchers in this field are very busy, so we think this is a useful way of writing for those who don't want to read the full text.

11. Page 7, Structural analysis using NMR: These experimental results must be more detailed discussed.

**** We have added a detailed discussion.

12. Page 7, Absorption from combined heavy metal cation/anion solutions: This section should be reorganized. All experimental results must be clearly presented and detailed discussed. Irrelevant general observations should be deleted.

**** We have moved the parts that should be written in the introduction to Introduction.

13. Page 10, Terpolymer hydrogel absorption swelling behavior: The same observations as above.

14. Page 13, Copolymer hydrogel absorption, swelling and effect of internal pH environment: The same observations.

**** Regarding points 13 and 14, We think it is necessary to show the results because they are not the same thing.

15. Page 17, Conclusion: This section must be reorganized. Delete (1), (2),…, and provide in this section a clear presentation of the most important experimental results and findings.

**** We intended to make the corrections as you told us to do.

16. Page 18, References: The number of references must be increased.

**** We added some references.

17. Fig. 8 should be moved into Supplementary materials.

**** We moved it to "Supporting information".

Dear Reviewer 2,

Thank you for your valuable feedback. The author who originally wrote the manuscript and the author who took over this have moved on for work, and I need time to communicate with them, so the revisions have been delayed, but I appreciate your continued support. In consideration of your comments, we have made as many corrections as possible. We thought it would be easier to understand if we wrote them in the email text we received, so we wrote our response after mark of ****. We have replied each item below.

Best regards,

Hirotaka Okabe and other authors

Reviewer #2: Dear Authors

Thank you for your interest in the PLOS ONE. I have received your manuscript (PONE-D-23-29103) entitled “Ionized acrylamide copolymer and terpolymer hydrogel absorbents for fractional recovery of positive and negative heavy metal ions from wastewater". However, after reading your paper, you should take the comments very carefully. My opinion is that the manuscript should be Reviewed again after Major revision and the manuscript should be resubmitted.

The following are comment to the author:-

1- Chemical structure of the used materials should be provided in experimental part.

**** We provide the chemical structures of the used materials in experimental part.

2- FTIR analysis should be performed for some selected prepared terpolymers and also for mono-polymers.

**** We provide the pre-liminary FT-IR data in Supporting Information.

3- All analytical devices should be mentioned separately in the experimental part with its manufacturing countries, model, and condition for each tool.

**** We tried rewriting them like you said.

4- In results and discussion there is no equation for the protonation of (SS) like that of AAm and (DMAPAA).

**** We added the equations in Materials.

5- References should be added for the protonation and dissociation of both AAm and (DMAPAA).

**** We added the reference.

6- All equations should be numbered.

**** We did so.

7- NMR not discussed in the manuscript, the authors should add brief discussion.

**** We added the discussion.

8- Figure captions should be added at the end of each provided figure.

**** We have added figure captions except where it is not necessary.

9- In electroplating industries as mentioned in introduction part, where the authors can apply their best hydrogel composition, how they can change the pH of the resulting actual wastewater?

**** We have written about this at the end of the text.

6. PLOS authors have the option to publish the peer review history of their article (what does this mean?). If published, this will include your full peer review and any attached files.

Do you want your identity to be public for this peer review? For information about this choice, including consent withdrawal, please see our Privacy Policy.

Reviewer #1: No

Reviewer #2: Yes: Prof. A. M. Abdel-Ghaffar

---

## [Editor Report · Decision Letter 1]

17 Jan 2024

Ionized acrylamide-based copolymer / terpolymer hydrogels for recovery of positive and negative heavy metal ions

PONE-D-23-29103R1

Dear Professor Okabe,,

We’re pleased to inform you that your manuscript has been judged scientifically suitable for publication and will be formally accepted for publication once it meets all outstanding technical requirements.

Kind regards,

Nayan Ranjan Singha, Ph.D.

Academic Editor

PLOS ONE
---

## [Editor Report · Acceptance letter]

21 Feb 2024

PONE-D-23-29103R1 

PLOS ONE

Dear Dr. Okabe, 

I'm pleased to inform you that your manuscript has been deemed suitable for publication in PLOS ONE. Congratulations! Your manuscript is now being handed over to our production team.

Kind regards, 

on behalf of

Dr. Nayan Ranjan Singha 

Academic Editor

PLOS ONE